# Roles of IL-7R Induced by Interactions between Cancer Cells and Macrophages in the Progression of Esophageal Squamous Cell Carcinoma

**DOI:** 10.3390/cancers15020394

**Published:** 2023-01-06

**Authors:** Yu Kitamura, Yu-ichiro Koma, Kohei Tanigawa, Shuichi Tsukamoto, Yuki Azumi, Shoji Miyako, Satoshi Urakami, Takayuki Kodama, Mari Nishio, Manabu Shigeoka, Yoshihiro Kakeji, Hiroshi Yokozaki

**Affiliations:** 1Division of Pathology, Department of Pathology, Kobe University Graduate School of Medicine, Kobe 650-0017, Japan; 2Division of Gastro-Intestinal Surgery, Department of Surgery, Kobe University Graduate School of Medicine, Kobe 650-0017, Japan; 3Division of Gastroenterology, Department of Internal Medicine, Kobe University Graduate School of Medicine, Kobe 650-0017, Japan

**Keywords:** esophageal cancer, tumor-associated macrophages, direct co-culture, IL-7R, prognostic factor, immunohistochemistry

## Abstract

**Simple Summary:**

Tumor-associated macrophages (TAMs) play significant roles in the progression of numerous types of cancers. We previously reported that the density of TAM infiltration in esophageal squamous cell carcinoma (ESCC) tissues is correlated with poor prognosis in patients with ESCC. To elucidate the significance of the direct interaction in the ESCC microenvironment, we previously established a direct co-culture assay of ESCC cells and macrophages. In the present study, direct co-culture of ESCC cells with macrophages induced interleukin 7 receptor (IL-7R) expression in ESCC. IL-7R overexpression promoted ESCC cell survival and growth via the activation of the Akt and Erk1/2 signaling pathways. Additionally, the IL-7/IL-7R axis promoted ESCC cell migration via the Akt and Erk1/2 signaling pathways. ESCC patients with high IL-7R expression in cancer nests exhibited poor disease-free survival. IL-7R could be a novel therapeutic target for ESCC.

**Abstract:**

High infiltration of tumor-associated macrophages (TAMs), which contribute to the progression of several cancer types, is correlated with poor prognosis of esophageal squamous cell carcinoma (ESCC). In addition to the previously reported increase in migration and invasion, ESCC cells co-cultured directly with macrophages exhibited enhanced survival and growth. Furthermore, interleukin-related molecules are associated with ESCC; however, the precise mechanism underlying this association is unclear. Therefore, we explored the role of interleukin-related molecules in ESCC progression. A cDNA microarray analysis of monocultured and co-cultured ESCC cells revealed that the interleukin 7 receptor (IL-7R) was upregulated in ESCC cells co-cultured with macrophages. Overexpression of IL-7R promoted the survival and growth of ESCC cells by activating the Akt and Erk1/2 signaling pathways. The IL-7/IL-7R axis also contributed to the promotion of ESCC cell migration via the Akt and Erk1/2 signaling pathways. Furthermore, immunohistochemistry showed that ESCC patients with high IL-7R expression in cancer nests exhibited a trend toward poor prognosis in terms of disease-free survival, and showed significant correlation with increased numbers of infiltrating macrophages and cancer-associated fibroblasts. Therefore, IL-7R, which is upregulated when directly co-cultured with macrophages, may contribute to ESCC progression by promoting the development of various malignant phenotypes in cancer cells.

## 1. Introduction

Esophageal cancer is aggressive, and the sixth leading cause of cancer-related mortalities worldwide [1]. According to global cancer statistics, approximately 604,100 people had esophageal cancer in 2020, with approximately 544,000 associated deaths [1]. Esophageal cancer is histologically classified under two major subtypes: esophageal squamous cell carcinoma (ESCC) and esophageal adenocarcinoma [2]. ESCC is the predominant subtype in Asian countries, including Japan [3,4]. Despite treatment with surgery, chemotherapy, and radiotherapy, ESCC shows a poor prognosis with a 5-year overall survival rate of approximately 20% after diagnosis, mainly attributed to a high rate of distant metastasis and local recurrence caused by rapid progression [5,6,7].

Cancer tissues are composed of cancer cells and various stromal cells. The latter include macrophages, fibroblasts, and endothelial cells [8]. Cancer cells and stromal cells interact with each other within the tumor microenvironment (TME), the environment surrounding cancer cells [9,10]. Macrophages are major components of cancer stroma, and those within the TME are referred to as tumor-associated macrophages (TAMs) [8,11]. There are two main macrophage phenotypes: classically activated macrophages (M1) and alternatively activated macrophages (M2) [11,12]. Although the M1/M2 classification is widely accepted, macrophages in actual cancer patients often exhibit a mixture of M1/M2 characteristics, and the use of terms such as “M1-like” or “M2-like” has recently been considered more suitable [13]. Interleukin (IL)-4, IL-10, and IL-13 induce M2-like macrophages [14]. TAMs exhibit M2-like phenotypes, such as anti-inflammatory and immunosuppressive phenotypes, in many types of cancer [15,16]. TAMs also secrete various factors that facilitate cell survival, growth, migration, and invasion in various cancers [17,18]. In fact, we previously reported that the TAM infiltration density in ESCC tissue is correlated with poor prognosis among patients with ESCC [19].

To investigate the interaction between cancer cells and TAMs, various humoral factors and their mediators have been studied via indirect co-culture assays between ESCC cells and TAMs. In addition, images of cancer cells adhered to TAMs within cancer tissues have been observed [20]. We previously established a direct co-culture assay between ESCC cells and macrophages derived from human peripheral blood monocytes to elucidate their direct interactions in the TME. Direct co-culture with macrophages promoted the migration and invasion abilities of ESCC cells. We then conducted cDNA microarray analysis to investigate gene expression changes caused by co-culture with macrophages. Among the upregulated genes, *S100A8* and *S100A9* were associated with poor prognosis in patients with ESCC and enhanced the migration and invasion abilities of ESCC cells via the Akt and p38 MAPK signaling pathways [20].

Interleukin-related molecules are associated with ESCC [21,22,23]; however, the precise mechanism underlying their role in ESCC progression is unclear. Therefore, in the present study, we focused on the upregulation of interleukin-related genes in ESCC cells by directly co-culturing them with macrophages (Appendix A). As plasma membrane receptors represent potential therapeutic targets, we investigated the role of interleukin 7 receptor (IL-7R) because its expression was the most upregulated among interleukin-related molecules in ESCC cells co-cultured with macrophages. The results of the present study provide insights into potential therapeutic targets and prognostic factors for ESCC.

## 2. Materials and Methods

### 2.1. Cell Lines and Cell Cultures

The ESCC cell lines, TE-9, -10, and -11, were purchased from the RIKEN BioResource Center (Tsukuba, Japan). The original tumors from which TE-9, -10, and -11 cells were derived, were poorly, well, and moderately differentiated ESCC, respectively. Cell culture was performed according to a previously reported method [20]. All three ESCC cell lines were cultivated in RPMI-1640 (FUJIFILM Wako Pure Chemical, Osaka, Japan) with 10% fetal bovine serum (FBS; Sigma-Aldrich, St. Louis, MO, USA) and 1% antibiotic-antimycotic (Thermo Fisher Scientific, Waltham, MA, USA) at 37 °C under 5% CO_2_.

### 2.2. Preparation of Macrophages Derived from Peripheral Blood Monocytes

Macrophages were generated using a previously reported method [20]. To isolate CD14^+^ peripheral blood monocytes (PBMos), human peripheral blood samples from healthy donors were labeled with anti-CD14 microbeads (#130-050-201, Miltenyi Biotec, Bergisch Gladbach, Germany) and separated by positive selection using an autoMACS Pro Separator (Miltenyi Biotec). Subsequently, 1 × 10^6^ PBMos were seeded in RPMI-1640 with 10% FBS and 25 ng/mL recombinant human macrophage-colony stimulating factor (#216-MC; R&D Systems, Minneapolis, MN, USA) in a 10-cm dish and cultivated for 6 days.

### 2.3. Direct Co-Culture Assay of ESCC Cells and Macrophages

A direct co-culture assay was performed using a previously reported method [20]. First, the established macrophages were washed three times with serum-free RPMI-1640. Then, macrophages were incubated with 2 × 10^6^ TE-9, -10, or -11 cells in serum-free RPMI-1640 for 2 days. The co-cultured dish was then washed with phosphate-buffered saline (Nacalai Tesque, Kyoto, Japan) and treated with trypsin-EDTA (Nacalai Tesque). Finally, co-cultured TE-9, -10, or -11 cells were labeled with anti-EpCAM microbeads (#130-061-101, Miltenyi Biotec) and isolated by positive selection using an autoMACS Pro Separator (Miltenyi Biotec). Subsequently, 2 × 10^6^ TE-9, -10, or -11 cells were seeded in separate plates and isolated by positive selection as monocultured controls, using a method similar to that described for co-cultured ESCC cells.

### 2.4. Reverse Transcription Polymerase Chain Reaction (RT-PCR) and Quantitative Real-Time Polymerase Chain Reaction (qRT-PCR)

Total RNA was extracted from cultured cells using the RNeasy Mini Kit (Qiagen, Hilden, Germany) according to the manufacturer’s instructions. The concentration of RNA was measured using a NanoDrop Lite (Thermo Fisher Scientific). RT-PCR and qRT-PCR were performed to assess the expression levels of interleukin-7 receptor (*IL7R*) and glyceraldehyde-3-phosphate dehydrogenase (*GAPDH*) mRNA. RT-PCR products were detected by 2% agarose gel electrophoresis. qRT-PCR was performed using SYBR Green PCR Master Mix (Applied Biosystems, Foster City, CA, USA) and detected using an ABI StepOnePlus Real-Time PCR System (Applied Biosystems). The qRT-PCR was performed based on the comparative threshold cycle (Ct) method, as described in previous studies [19,20]. The Ct values were normalized to that of *GAPDH*. The primer sequences used were as follows: *IL7R* (RT-PCR, qRT-PCR), 5′-ACG ATG TAG CTT ACC GCC AGG AAA-3′ (forward), 5′-TCT CTG CAG GAG TGT CAG CTT TGT-3′ (reverse); *GAPDH* (RT-PCR), 5′-ACC ACA GTC CAT GCC ATC AC-3′ (forward), 5′-TCC ACC ACC CTG TTG CTG TA-3′ (reverse); *GAPDH* (qRT-PCR), 5′-GCA CCG TCA AGG CTG AGA AT-3′ (forward), 5′-ATG GTG GTC AAG ACG CCA GT-3′ (reverse).

### 2.5. Western Blot Analysis

Protein extraction was performed via a method reported in a previous study [20]. Total cellular protein was extracted by a lysis buffer (50 mM Tris-HCl at pH 7.5, 125 mM NaCl, 5 mM EDTA, and 0.1% Triton X-100) with 1% protease/phosphatase inhibitor cocktails (Sigma-Aldrich). The concentration of protein was quantified using a NanoDrop Lite (Thermo Fisher Scientific). Protein samples were separated on a 5–20% gradient sodium dodecyl sulfate-polyacrylamide gel, then transferred to a polyvinylidene difluoride membrane using an iBlot2 system (Invitrogen, Carlsbad, CA, USA). After blocking with 5% skim milk, the membrane was treated with primary antibodies for 24–48 h at 4 °C. The membrane was then treated with secondary antibody for 90 min at room temperature. ImmunoStar reagents (FUJIFILM Wako Pure Chemical) were used for the detection of protein bands.

The primary antibodies used were as follows: mouse monoclonal antibody against IL-7R (#sc-514445, Santa Cruz Biotechnology, Dallas, TX, USA) and rabbit monoclonal antibodies against phosphorylated (p) Akt (Ser473) (#4060, Cell Signaling Technology; CST, Beverly, MA), pAkt (Thr308) (#2965, CST), total Akt (#9272, CST), pErk1/2 (Thr202/Tyr204) (#4370, CST), total Erk1/2 (#9102, CST), E-cadherin (#3195, CST), N-cadherin (#4061, CST), vimentin (#5741, CST), pRIP (Ser166) (#65746, CST), RIP (#3493, CST), pMLKL (Ser358) (#91689, CST), MLKL (#14993, CST), cleaved Caspase-3 (Asp175) (#9664, CST), Caspase-3 (#14220, CST), cleaved Caspase-8 (Asp384) (#9748, CST), Caspase-8 (#4790, CST), Cyclin D1 (#2926, CST), and β-actin (#4970, CST). The secondary antibodies were horseradish peroxidase (HRP)-conjugated anti-mouse IgG (#NA931V; GE Healthcare, Chicago, IL, USA) and HRP-conjugated anti-rabbit IgG (#NA934V; GE Healthcare).

### 2.6. Overexpression of IL-7R in ESCC Cell Lines

The expression plasmid pCMV6-IL7R (#RC209687, OriGene Technologies, Rockville, MD, USA), which encodes the *IL7R* ORF, was employed to overexpress IL-7R in the ESCC cell lines. Lipofectamine^®^ 3000 reagent (Thermo Fisher Scientific) was used to transfect plasmid DNA into ESCC cells. Following transfection for 2 days, RPMI-1640 containing 10% FBS with 300 or 100 µg/mL geneticin (G418, Thermo Fisher Scientific) was used to select stably transfected TE-9 and -11 cells, respectively. ESCC cells transfected with empty pCMV6-Entry vectors (#PS100001, OriGene Technologies) were used as negative controls (control). The limiting dilution method was continued for at least a month, and colonies of cloned cells were obtained. One control and two IL-7R-overexpressing clones (OE IL-7R) were selected for TE-9 and -11 cells.

### 2.7. IL-7R Knockdown in ESCC Cell Lines Using Small Interfering (si) RNA

Transfection of siRNA was performed according to a previously reported method [20]. TE-9, -10, and -11 cells were transfected with 20 nM siRNA against IL-7R (siIL-7R; #sc-35664; Santa Cruz Biotechnology) or negative control siRNA (siNC; Sigma-Aldrich) by Lipofectamine RNAiMAX (Invitrogen) and cultivated for 72 h.

### 2.8. Cell Proliferation Assay

Survival and growth assays for ESCC cells were performed in 96-well plates. For the survival assay, 1 × 10^4^ cells/well were seeded in serum-free RPMI-1640 medium. For the growth assay, 5 × 10^3^ cells/well were seeded in RPMI-1640 with 1% FBS. In both assays, the cells were incubated at 37 °C under 5% CO_2_ for 48 h. In certain experiments, the ESCC cells were treated with/without 20 ng/mL recombinant human IL-7 (rhIL-7, #207-IL; R&D Systems). Following the addition of CellTiter 96 AQueous One Solution Reagent (20 µL/well; Promega, Madison, WI, USA), the absorbance was measured at 492 nm using an Infinite 200 PRO microplate reader (Infinite 200 PRO; Tecan, Mannedorf, Switzerland). A schematic of the experimental procedure of the cell proliferation assay is shown in Appendix A.

### 2.9. Transwell Migration Assay

The transwell migration assay for ESCC cells was performed in 24-well plates using RPMI-1640 containing 0.1% FBS (800 μL/well) with/without rhIL-7 (20 ng/mL). The cell culture insert (pore size, 8-μm; BD Falcon, Lincoln Park, NY, USA) was used as the upper chamber with 1 × 10^5^ cells in 300 μL RPMI-1640 containing 0.1% FBS. In certain experiments, the ESCC cells were treated with either PI3K inhibitor (LY294002, 10 μM; CST) or MEK1/2 inhibitor (PD98059, 10 μM; CST), and either a neutralizing antibody against IL-7R (1 μg/mL; #sc-514445, Santa Cruz Biotechnology) or normal mouse IgG (1 μg/mL; #sc-2025, Santa Cruz Biotechnology) as the negative control in the presence of rhIL-7 (20 ng/mL). After 48 h of incubation, the membrane of the upper chamber was stained using Diff-Quik (Sysmex, Kobe, Japan), and the cells that migrated to the underside of the membrane were enumerated as described in a previous study [20]. A schematic of the experimental procedure of the transwell migration assay is shown in Appendix A.

### 2.10. Wound Healing Assay

ESCC cells in RPMI-1640 with 10% FBS were seeded in 24-well plates (600 μL/well). The ESCC cells were incubated for up to 24 h until 100% confluency was achieved; thereafter, wounds were prepared using 1000-µL pipette tips. After washing with phosphate-buffered saline, wounded areas were captured using a CCD camera (40× magnification; Olympus, Tokyo, Japan). The medium was then changed to serum-free RPMI-1640 with/without rhIL-7 (20 ng/mL). In certain experiments, the ESCC cells were treated with either 10 μM LY294002 or 10 μM PD98059, and either a neutralizing antibody against IL-7R (1 μg/mL) or normal mouse IgG (1 μg/mL) as the negative control in the presence of rhIL-7 (20 ng/mL). After incubation for 24 h (for TE-11) or 48 h (for TE-9 and -10), each well was washed and the cells migrating to the wounded area were captured using a CCD camera (Olympus). We evaluated the migration ability by measuring the cell-free area with the polygon selection tool of ImageJ v1.52v (National Institutes of Health, Bethesda, MD, USA). Furthermore, the percentage wound coverage was compared between before treatment (A_before_) and after 24 or 48 h (A_after_); [(A_before_ − A_after_)/A_before_] × 100. A schematic of the experimental procedure of the wound healing assay is shown in Appendix A.

### 2.11. Enzyme-Linked Immunosorbent Assay (ELISA)

Macrophages, and TE-9, -10, and -11 cells (1 × 10^6^ cells/well) in RPMI-1640 with 10% FBS were cultured in 6-well plates for 48 h. In addition, after separation using an autoMACS Pro Separator (Miltenyi Biotec), monocultured and co-cultured TE-9, -10, and -11 cells (3 × 10^5^ cells/well) in RPMI-1640 with 10% FBS were cultured in 6-well plates for 48 h. The cell culture supernatants were collected via a previously described method [19]. The concentrations of IL-7 in the cell culture supernatants were measured using the Human IL-7 Quantikine HS ELISA Kit (#HS750; R&D Systems) according to the manufacturer’s instructions. The optimal density of each well was determined at 490 nm using an Infinite 200 PRO microplate reader.

### 2.12. Tissue Samples

ESCC patients who underwent surgical resection at Kobe University Hospital (Kobe, Japan) between 2005 and 2010 were selected. Exclusion criteria were patients who received neoadjuvant therapy (chemotherapy and/or radiotherapy) before resection. In all, 69 human ESCC tissue samples were registered in the present study. Clinicopathological and histological factors were classified according to the Japanese Classification of Esophageal Cancer [24] and the Union for International Cancer Control TNM classification [25]. The study protocol was approved by the Institutional Review Board of Kobe University (B210103) and was conducted in accordance with the guidelines of the 1964 Declaration of Helsinki. All patients gave informed consent for the use of tissue samples and clinical data.

### 2.13. Immunohistochemistry

Immunohistochemistry analyses for 4-μm-thick tissue sections were performed using the BOND Polymer Refine Detection Kit (Leica Biosystems, Bannockburn, IL, USA) on the BOND-MAX automated system (Leica Biosystems). The dilution ratio of the antibody was set at 1:500. Rabbit IL-7R antibody (#CSB-PA011670EA01HU, Cusabio Technology, Houston, TX, USA) was adopted as the primary antibody, and normal rabbit polyclonal IgG (#PM035, Medical & Biological Laboratories; MBL, Nagoya, Japan) was used as a negative control (Appendix A). The staining intensity in cancer nests was scored as follows: 0 indicated no staining; 1 indicated weaker than normal esophageal epithelium; and 2 indicated equal to or stronger than normal esophageal epithelium. Cases with scores of 0 and 1 were classified as the low-expression group, and cases with scores of 2 were classified as the high-expression group. All tissue samples were evaluated by two expert pathologists (Y.-i.K. and H.Y.) and one surgeon (Y.K. [Yu Kitamura]), based on the above-mentioned criteria.

### 2.14. Statistical Analyses

All in vitro experiments were performed in triplicate and repeated independently three times. Significant differences between conditions were determined using a two-tailed Student’s *t*-test, and the data were presented as mean ± standard error of mean (SEM). The relationship between immunohistochemical results and clinicopathological factors was analyzed using the *χ*^2^ test. The overall, disease-free, and cause-specific survival curves were estimated using the Kaplan–Meier method and evaluated using a log-rank test. Statistical significance was set at *p* < 0.05. Statistical analyses were conducted using IBM SPSS Statistics 22 (IBM Corp., Armonk, NY, USA).

## 3. Results

### 3.1. Direct Co-Culture with Macrophages Promotes the Survival and Growth of ESCC Cells through the Activation of Akt and Erk1/2 Signaling

Direct co-culture with macrophages could enhance ESCC cell survival and growth, in addition to the previously reported promotion of the migration and invasion abilities of ESCC cells. Co-cultured TE-9, -10, and -11 cells increased cell survival significantly compared with that in monocultured TE-9, -10, and -11 cells, respectively (Figure 1A). Co-cultured TE-10 and -11 cells exhibited significantly enhanced growth compared with monocultured TE-10 and -11 cells, respectively (Figure 1B). Co-cultured TE-9 cells grew at a rate that was relatively similar to that of monocultured TE-9 cells (Figure 1B).

The signaling pathways activated in ESCC cells following direct co-culture with macrophages were then investigated using Western blot analysis. Direct co-culture with macrophages increased the phosphorylation levels of Erk1/2 in TE-9, -10, and -11 cells, in addition to the previously reported enhancement of Akt phosphorylation (Figure 1C and Appendix A).

### 3.2. IL-7R Expression Is Upregulated in ESCC Cells following Direct Co-Culture with Macrophages

We performed qRT-PCR to confirm that direct co-culture with macrophages increased the expression of *IL7R* mRNA in TE-9, -10, and -11 cells (Figure 1D). Thereafter, we examined the expression levels of IL-7R protein in ESCC cell lines. Western blotting results revealed that TE-10 and -11 cells showed upregulated IL-7R expression upon direct co-culture with macrophages; however, TE-9 cells did not exhibit such a change (Figure 1E and Appendix A).

### 3.3. IL-7R Overexpression in ESCC Cells Markedly Promotes Cell Survival and Growth

To investigate the significance of high IL-7R expression in ESCC cells, we overexpressed IL-7R in ESCC cells via gene transfection. TE-9 cells were selected for transfection because IL-7R protein was not upregulated and growth was not promoted in TE-9 cells directly co-cultured with macrophages. TE-9 cells stably transfected with the IL-7R expression vector (OE IL-7R#1 and OE IL-7R#2) had significantly higher IL-7R mRNA and protein expression than those transfected with the control vector (control) (Figure 2A–C and Appendix A). To investigate the effect of IL-7R overexpression on survival, growth, and cell signaling, TE-9 (OE IL-7R#1 and OE IL-7R#2) and TE-9 (control) were compared. Compared with TE-9 (control), both IL-7R-overexpressing clones (OE IL-7R#1 and OE IL-7R#2) exhibited significantly increased survival, growth, and phosphorylation levels of Akt and Erk1/2 (Figure 2D–F and Appendix A). TE-11 cells also showed similar results with regard to the effect of IL-7R overexpression (Appendix A).

### 3.4. Knockdown of IL-7R in ESCC Cells Markedly Suppresses Cell Survival and Growth

To investigate the association between IL-7R expression and various phenotypes of ESCC cells, IL-7R was knocked down in ESCC cells using siRNA. The expression levels of *IL7R* mRNA and IL-7R protein in TE-9, -10, and -11 cells transfected with siIL-7R were lower than those in cells transfected with siNC (Figure 3A–C and Appendix A). Cell proliferation assays demonstrated that the knockdown of IL-7R in TE-9, -10, and -11 cells suppressed survival (Figure 3D) and growth (Figure 3E) significantly compared with those in siNC-transfected TE-9, -10, and -11 cells, respectively. Transwell migration and wound healing assays were performed using TE-9, -10, and -11 cells transfected with siIL-7R; however, cell migration ability could not be assessed as IL-7R knockdown reduced cell survival and growth significantly.

### 3.5. Exogeneous IL-7 Promotes ESCC Cell Migration Ability via the Akt and Erk1/2 Signaling Pathways

As IL-7 is a specific ligand of IL-7R, additional assays were performed to investigate the effect of exogeneous IL-7 on ESCC cells using recombinant human IL-7 (rhIL-7). Notably, rhIL-7 had no effect on cell survival (Figure 4A) or growth (Figure 4B), but it enhanced the migration abilities of TE-9, -10, and -11 cells significantly in the transwell migration assay (Figure 4C) and wound healing assay (Figure 4E). Moreover, the anti-IL-7R neutralizing antibody significantly suppressed rhIL-7-induced migration of TE-10 and -11 cells; however, there was no significant difference in TE-9 cells in the transwell migration assay (Figure 4D). Furthermore, the anti-IL-7R neutralizing antibody inhibited the rhIL-7-induced migration of TE-9, -10, and -11 cells significantly in the wound healing assay (Figure 4F).

Western blot analyses were used to investigate changes in cell signaling induced by exogenous IL-7. Phosphorylated Akt and Erk1/2 levels were upregulated in TE-9, -10, and -11 cells within 1 h of rhIL-7 treatment (Figure 5A and Appendix A). However, rhIL-7 treatment for 48 h had no effect on the expression levels of epithelial-mesenchymal transition (EMT)-related factors in TE-9, -10, or -11 cells (Appendix A). Furthermore, migration assays were carried out to evaluate whether Akt and Erk1/2 signaling contributes to the rhIL-7-induced migration of TE-9, -10, and -11 cells. The rhIL-7-induced migration of TE-9, -10, and -11 cells was suppressed significantly by a PI3K inhibitor or a MEK1/2 inhibitor in the transwell migration assay (Figure 5B) and in the wound healing assay (Figure 5C). The IL-7/IL-7R axis promotes the migration of ESCC cells via the Akt and Erk1/2 pathways.

### 3.6. High Expression of IL-7R in Cancer Nests Is Associated with the Infiltration Levels of Tumor-Associated Macrophages (TAMs) or Cancer-Associated Fibroblasts (CAFs) and Tends to Be Correlated with Poor Prognosis in ESCC Patients

Finally, we evaluated the expression levels of IL-7R in human ESCC tissue samples and examined their association with patient prognoses and clinicopathological factors. To this end, we performed immunohistochemistry for IL-7R in 69 human ESCC tissue samples, which were divided into two groups based on the staining intensity: low or high IL-7R expression (Figure 6A). Kaplan–Meier analysis showed that ESCC patients with high IL-7R expression demonstrated a higher tendency for poorer prognosis than those with low IL-7R expression in terms of disease-free survival (*p* = 0.055); however, there was no correlation between IL-7R expression and overall survival (*p* = 0.512) or cause-specific survival (*p* = 0.394) (Figure 6B).

Furthermore, we explored the association between IL-7R expression and clinicopathological factors in ESCC tissue samples. Although clinicopathological factors were not significantly associated with high IL-7R expression levels, histological grade (*p* = 0.088), depth of tumor invasion (*p* = 0.075), lymphatic vessel invasion (*p* = 0.081), blood vessel invasion (*p* = 0.053), and lymph node metastasis (*p* = 0.053) tended to be correlated with IL-7R expression in ESCC patients (Table 1). Moreover, high IL-7R expression was associated with a higher number of infiltrating CD68-, CD163-, and CD204-positive macrophages (*p* = 0.041, *p* = 0.028, and *p* = 0.006, respectively). Notably, high levels of expression of α-smooth muscle actin (αSMA) (*p* = 0.015) and fibroblast activation protein (FAP) (*p* = 0.048), which are known CAF markers, were also significantly associated with high IL-7R expression in cancer nests (Table 1).

## 4. Discussion

We previously demonstrated that the direct co-culture of ESCC cells with macrophages promotes cell migration and invasion [20]. The present study demonstrates that direct co-culture with macrophages improves ESCC cell survival and growth. Previous studies reported that interactions between cancer cells and macrophages through humoral factors contribute to the promotion of cell growth in several cancers [27,28,29,30]. We also reported that ESCC cells indirectly co-cultured with macrophages exhibit enhanced cell growth [31,32]. Furthermore, direct contact between tumor cells and peripheral blood-derived macrophages reportedly promotes growth of glioma [33] and sarcoma [34] cells. However, the enhancement of ESCC cell proliferation through direct contact with macrophages is a novel finding. Elucidation of this phenotype may reveal novel mechanisms underlying ESCC progression.

IL-7R is a heterodimeric complex consisting of IL-7Rα and a common γ-chain; the latter is shared by IL-2, IL-4, IL-7, IL-9, IL-15, and IL-21 receptors. IL-7Rα is expressed in the lymphoid system. IL-7 is a 25-kDa secreted soluble globular protein encoded by the *IL7* gene and is a ligand for IL-7R. IL-7R is expressed in both naïve and memory T-cells, and IL-7 signaling is required for the long-term survival of all T-cell populations [35,36]. Several studies have found that IL-7R expression facilitates tumor progression; for example, high levels of IL-7R expression in lung and breast cancer tissues are associated with poor prognosis and tumor progression [37,38]. Meanwhile, *IL7R* has been described as a suppressor gene in hepatocellular carcinoma [39]. Therefore, the role of IL-7R may differ among cancer types. In silico analysis revealed that *IL7R* gene expression was higher in ESCC tissue samples than in normal esophageal tissue samples [40]. However, the biological role of IL-7R in the ESCC microenvironment is unclear. To the best of our knowledge, ours is the first study that reports the role of IL-7R in ESCC progression.

We demonstrated that TE-9 cells, which did not exhibit the induction of IL-7R expression after direct co-culture with macrophages, exhibited increased cell survival and growth after *IL7R* gene transfection without the addition of exogenous IL-7. Furthermore, knockdown of IL-7R in the three ESCC cell lines suppressed cell survival and growth significantly. Knockdown of IL-7R also reduces the growth of non-small cell lung cancer and hepatitis B virus-related hepatoma cells due to a decrease in the expression of cell cycle-related protein Cyclin D1 [41,42], and induces the apoptosis of non-small cell lung cancer via the downregulation of anti-apoptosis-related protein Bcl-2 [43]. To explore the mechanism underlying the reduction of the proliferative potential of ESCC cells by IL-7R silencing, the expression levels of apoptosis-related protein Caspase-3 and Caspase-8, necroptosis-related protein RIP and MLKL, and cell cycle-related protein Cyclin D1 were examined by Western blot analysis (Appendix A). The results showed that silencing of IL-7R induced the activation of RIP, which is associated with necroptosis, in TE-11 cells, and suppressed the expression of Cyclin D1, which is associated with the cell cycle, in TE-9 and -10 cells. With regard to other molecules, no meaningful results were obtained to explain decreasing proliferation. It is difficult to attribute the cause of the IL-7R silencing-induced reduction in the proliferative capacity of ESCC cells to one specific mechanism. Exogenous IL-7 promotes breast cancer growth via the JAK/STAT pathway [44] and lung cancer cell growth via the c-Fos/c-Jun pathway [37]. In contrast, in the present study, we demonstrated that exogenous IL-7 had no effect on tumor growth in ESCC cells. Similar results have been reported for prostate cancer [45,46], bladder cancer [47], and malignant pleural mesothelioma cells [48]. Therefore, we hypothesized that IL-7R expression activates the growth signal in ESCC. Although exogenous IL-7 did not affect the proliferative potential of ESCC cells, the influence of endogenous trace amounts of IL-7 (discussed later) cannot be ruled out. In addition, other ligands might have worked against IL-7R in an autocrine manner.

The present study reveals that the IL-7/IL-7R axis enhances cell migration via the activation of Akt and Erk1/2 signals in ESCC cells. The binding of IL-7 promotes heterodimerization of IL-7Rα and γc, leading to the activation of JAK1 and JAK3 and consequent downstream signaling, most notably of STAT1, STAT3, and STAT5, as well as the PI3K-Akt-mTOR and MEK-Erk pathways, in T-cell development and homeostasis [36,49]. The IL-7/IL-7R axis contributes to prostate cancer cell migration via the PI3K-Akt pathway [45,46]. Moreover, IL-7 promotes cell migration and invasion by upregulating Erk1/2-mediated MMP-9 expression in bladder cancer cells [47]. Herein, we found that the Akt and Erk1/2 pathways are both important for IL-7/IL-7R-induced migration in ESCC. Meanwhile, other researchers have shown that the IL-7/IL-7R axis contributes to EMT in prostate cancer cells [50] and breast cancer cells [51]. We also investigated the association between the IL-7/IL-7R axis and EMT; exogenous IL-7 did not alter the expression levels of EMT-related markers in ESCC cells. Such differences in responses to exogenous IL-7 indicate that the effect of IL-7 may vary depending on the origin and histological type of the cancer.

The association between IL-7R expression and patient prognosis has been investigated in breast cancer, pancreatic ductal adenocarcinoma, and lung cancer. Al-Rawi et al. reported that patients who died from breast cancer had significantly higher tumoral *IL7R* mRNA levels than those who survived [38]. Heo et al. indicated that the survival probability of pancreatic ductal adenocarcinoma patients with low IL-7R expression was modestly higher than that of patients with high IL-7R expression, as revealed by in silico analysis [52]. Ming et al. discovered that high IL-7R expression was significantly correlated with advanced stage, lymph node metastasis, and poor prognosis in terms of overall survival in lung cancer, based on immunohistochemistry [37]. Suzuki et al. also showed that IL-7R expression was associated with aggressive tumor features and frequent recurrence in clinical stage I lung adenocarcinoma by immunohistochemistry [53]. In addition, immunohistochemical expression of IL-7R in lung cancer patients was found to be an independent predictor of survival [54,55]. We also investigated the association between IL-7R expression and patient prognosis and clinicopathological factors in ESCC tissue samples using immunohistochemistry. Kaplan–Meier analysis showed that ESCC patients with high IL-7R expression demonstrated poorer disease-free survival than those with low IL-7R expression. Although no clinicopathological factors were significantly associated with high expression levels of IL-7R, histological grade, depth of tumor invasion, lymphatic invasion, vascular invasion, and lymph node metastasis tended to be related to the expression levels of IL-7R in patients with ESCC. However, a positive correlation was observed between the expression of IL-7R in ESCC cancer nests and the infiltration density of CD163- or CD204-positive cells, i.e., TAMs. Such findings support our hypothesis that IL-7R is upregulated in ESCC cells by direct contact with macrophages and promotes tumor growth along with other factors, such as GDF15 [31] and ANXA10 [32].

Notably, in the present study, the expression levels of αSMA and FAP, known as CAF markers, were positively correlated with the expression of IL-7R in ESCC tumor nests. IL-7 is produced in primary and secondary lymphoid organs and supports the development and maintenance of lymphocyte populations [36]. IL-7 expression has also been observed in human fibroblasts [49]. One of the possible precursors of CAFs is resident tissue fibroblasts [56]. Kröncke et al. reported the expression of *IL7* mRNA in primary cultured fibroblasts of the human foreskin [57]. In addition, in a mouse model, Boesch et al. reported that approximately 10% of the fibroblastic tumor stroma was occupied by IL-7-expressing CAFs in an orthotopic breast cancer tissue [58]. Furthermore, in a recent study, Yan et al. demonstrated that CAF-derived IL-7 promotes tumor progression in NOX5-positive ESCC cells [59]. Based on such reports, we hypothesized that IL-7 secreted by CAFs promotes ESCC cell migration. Despite considering IL-7 sources other than CAFs using ELISA, sufficient levels of IL-7 secretion could not be confirmed in ESCC cell lines or macrophages (Appendix A), or in monocultured or co-cultured ESCC cell lines (Appendix A).

The present study had several limitations. First, IL-7R-induced cell proliferation and migration in ESCC was not confirmed in vivo. However, a previous study reported that intratumoral ablation of IL-7-expressing fibroblasts impairs breast tumor growth and reduces the clonogenic potential of cancer cells in a mouse model [58]. Hence, ESCC xenograft mice should be used to further investigate the role of IL-7R in ESCC development. In addition, this study included 69 human ESCC tissue samples, which is considered a small case analysis. More cases could help verify the association between IL-7R expression and prognosis or clinicopathological factors. Furthermore, considering the strong effect of IL-7R expression on cell proliferation, the migration ability of ESCC cells transfected with siIL-7R could not be accurately assessed.

## 5. Conclusions

The present study demonstrates that IL-7R expression is upregulated in ESCC cells co-cultured directly with macrophages, and that high IL-7R expression promotes the survival and growth of cancer cells via activation of the Akt and Erk1/2 signaling pathways. Furthermore, the IL-7/IL-7R axis enhances the migration ability of ESCC cells via the activation of the Akt and Erk1/2 signaling pathways. High IL-7R expression in tumor nests was associated with the levels of TAM and CAF infiltration and tended to be associated with poor prognosis in patients with ESCC. Our findings show that IL-7R is a potential therapeutic target and prognostic factor for ESCC progression.

## Figures and Tables

**Figure 1 cancers-15-00394-f001:**
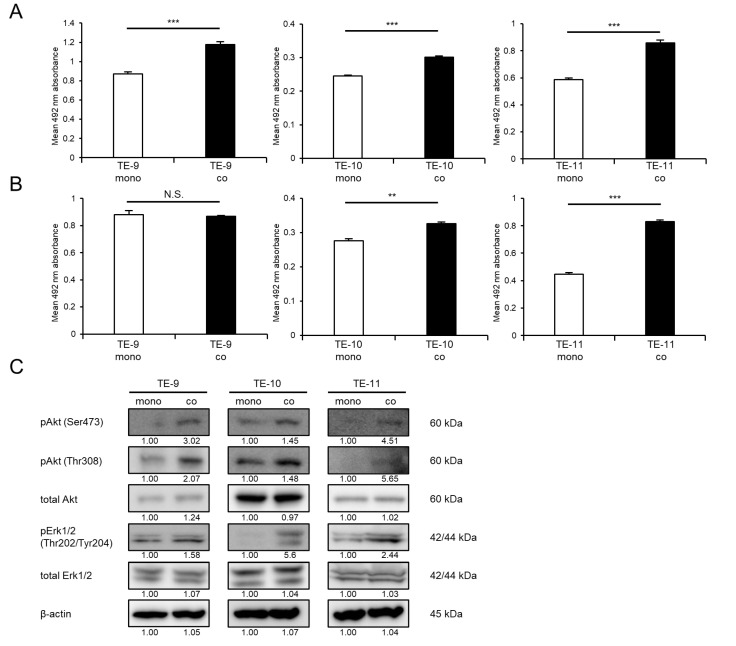
Effects of direct co-culture with macrophages on ESCC cell lines. ESCC cells were directly co-cultured with macrophages for 2 days and then separated from macrophages. (**A**,**B**) Survival assays (**A**) and growth assays (**B**) were performed to investigate the effects of direct co-culture. The three graphs compare the survival and growth of TE-9, -10, and -11 cells co-cultured with macrophages with those of monocultured TE-9, -10, and -11 cells, respectively. (**C**) Western blot analyses were performed to investigate the signaling pathways activated in TE-9, -10, and -11 cells by direct co-culture with macrophages. β-actin was used as an internal control. The expression level was normalized using ImageJ software, and the relative value was set as 1.00 for monocultured ESCC cells. (**D**) The expression levels of *IL7R* mRNA in TE-9, -10, and -11 cells monocultured and co-cultured with macrophages were confirmed by qRT-PCR. *GAPDH* was used as an internal control. (**E**) The expression levels of IL-7R protein in TE-9, -10, and -11 cells monocultured and co-cultured with macrophages were confirmed by Western blot analyses. β-actin was used as an internal control. The expression level was normalized using ImageJ software, and the relative value was set to 1.00 for monocultured ESCC cells. Data are presented as the mean ± SEM. N.S., not significant. ** *p* < 0.01, *** *p* < 0.001.

**Figure 2 cancers-15-00394-f002:**
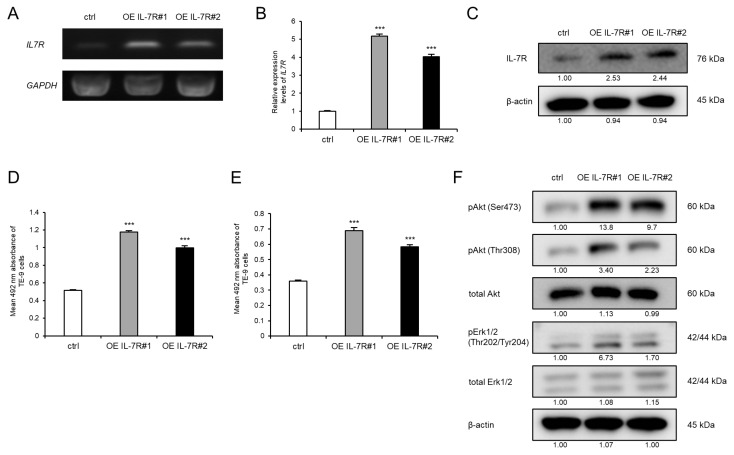
Overexpression of IL-7R promotes the survival and growth of TE-9 ESCC cells. (**A**,**B**) The expression levels of *IL7R* mRNA in TE-9 cells transfected with control vector (ctrl) or IL-7R expression vector (OE IL-7R) were confirmed by RT-PCR (**A**) and qRT-PCR (**B**). *GAPDH* was used as an internal control. (**C**) The expression levels of IL-7R protein in ctrl- and OE IL-7R-transfected TE-9 cells were confirmed by Western blot analyses. β-actin was used as an internal control. The expression level was normalized using ImageJ software; the relative value was set to 1.00 for ctrl-transfected cells. (**D**,**E**) Overexpression of IL-7R induced the survival (**D**) and growth (**E**) of TE-9 cells. (**F**) Western blot analyses demonstrated that the phosphorylation of Akt and Erk1/2 was induced by the overexpression of IL-7R in TE-9 cells. β-actin was used as an internal control. The expression level was normalized using ImageJ software; the relative value was set as 1.00 for ctrl-transfected cells. Data are presented as the mean ± SEM. *** *p* < 0.001.

**Figure 3 cancers-15-00394-f003:**
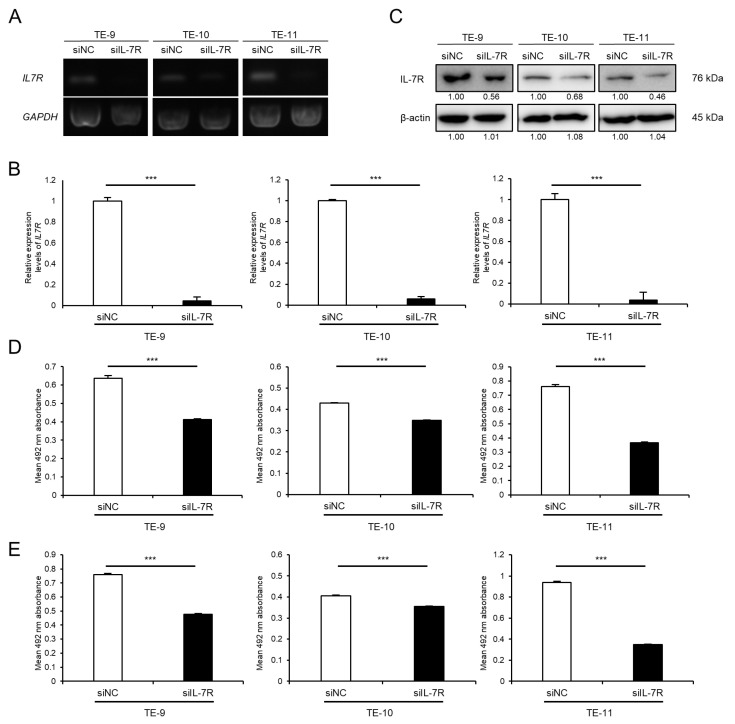
Knockdown of IL-7R suppresses the survival and growth of ESCC cells. (**A**,**B**) The expression levels of *IL7R* mRNA in TE-9, -10, and -11 cells transfected with 20 nM siRNA against IL-7R (siIL-7R) or negative control siRNA (siNC) were confirmed by RT-PCR (**A**) and qRT-PCR (**B**). *GAPDH* was used as an internal control. (**C**) The expression levels of IL-7R protein in TE-9, -10, and -11 cells transfected with siIL-7R or siNC, were confirmed by Western blot analyses. β-actin was used as an internal control. The expression level was normalized using ImageJ software; the relative value was set to 1.00 for siNC-transfected cells. (**D**,**E**) Cell proliferation assays were conducted to investigate the effect of IL-7R knockdown on the survival (**D**) and growth (**E**) of ESCC cells. Data are presented as the mean ± SEM. *** *p* < 0.001.

**Figure 4 cancers-15-00394-f004:**
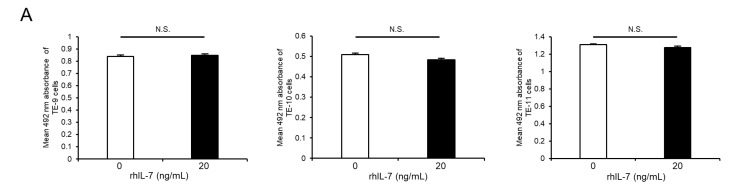
Exogenous IL-7 enhances the migration of ESCC cells via IL-7R without affecting cell survival or growth. (**A**,**B**) Cell proliferation assays were performed to investigate the effect of recombinant human IL-7 (rhIL-7, 20 ng/mL) on the survival (**A**) and growth (**B**) of TE-9, -10, and -11 cells. (**C**) Transwell migration assays were performed to investigate the effect of rhIL-7 on the vertical migration abilities of TE-9, -10, and -11 cells. The migrated cells were enumerated in four randomly selected areas. (**D**) Transwell migration assays were performed to investigate the effect of anti-IL-7R neutralizing antibodies on rhIL-7-induced vertical migration abilities of TE-9, -10, and -11 cells. Normal mouse IgG was used as a negative control. The migrated cells were counted in four randomly selected areas. (**E**) Wound healing assays were performed to investigate the effect of rhIL-7 on the horizontal migration abilities of TE-9, -10, and -11 cells. After incubation for a specific duration, the percentage wound coverage was measured and compared. (**F**) Wound healing assays were performed to investigate the effect of anti-IL-7R neutralizing antibodies on rhIL-7-induced horizontal migration abilities of TE-9, -10, and -11 cells. Normal mouse IgG was used as a negative control. After incubation for a specific duration, the percentage wound coverage was measured and compared. Data are presented as the mean ± SEM. N.S., not significant. * *p* < 0.05, ** *p* < 0.01, *** *p* < 0.001.

**Figure 5 cancers-15-00394-f005:**
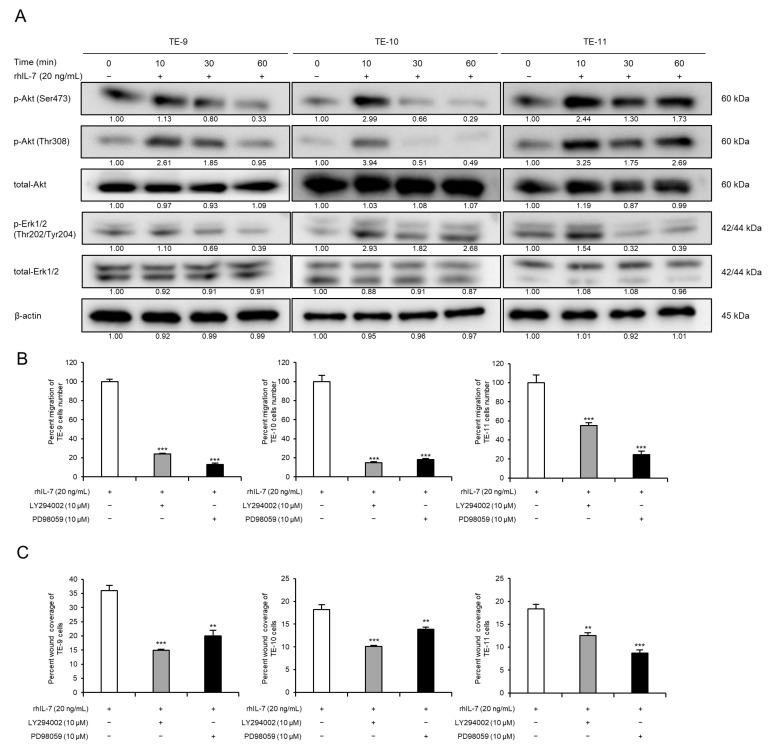
IL-7/IL-7R axis promotes migration of ESCC cells via Akt and Erk1/2 activation. (**A**) Western blot analyses demonstrated phosphorylated or total Akt and Erk1/2 levels in TE-9, -10, and -11 cells. Under serum-free conditions, TE-9, -10, and -11 cells were treated with recombinant human IL-7 (rhIL-7, 20 ng/mL) for 0, 10, 30, and 60 min. β-actin was used as an internal control. The expression level was normalized using ImageJ software; the relative value was set as 1.00 for rhIL-7-untreated cells. (**B**) Transwell migration assays were performed to investigate the effect of PI3K or MEK1/2 inhibitors on the migration of rhIL-7-treated ESCC cells. Transwell migration assays of TE-9, -10, and -11 cells treated with rhIL-7 (20 ng/mL) and inhibitors against PI3K (LY294002, 10 µM) or MEK1/2 (PD98059, 10 µM) were performed. The number of migrated cells was counted in four randomly selected areas. (**C**) Wound healing assays were performed to investigate the effect of PI3K or MEK1/2 inhibitors on the migration of rhIL-7-treated ESCC cells. Wound healing assays of TE-9, -10, and -11 cells treated with rhIL-7 (20 ng/mL) and inhibitors against PI3K (LY294002, 10 µM) or MEK1/2 (PD98059, 10 µM) were performed. After incubation for a specific duration, the percentage wound coverage was measured and compared. Data are presented as the mean ± SEM. N.S., not significant. ** *p* < 0.01, *** *p* < 0.001.

**Figure 6 cancers-15-00394-f006:**
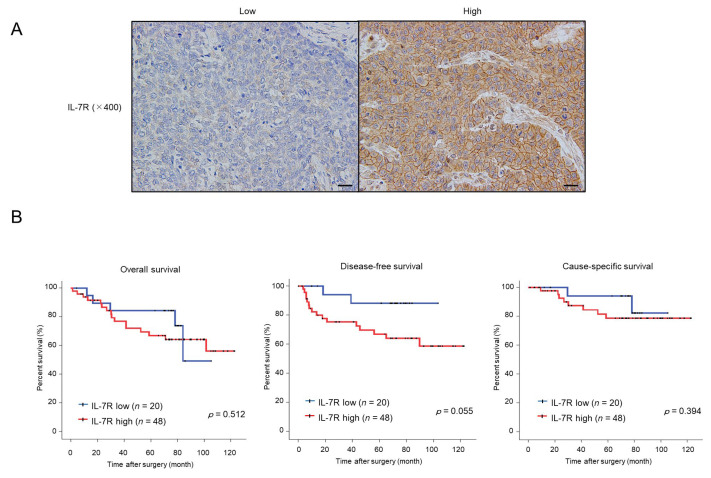
ESCC patients with tumors expressing high levels of IL-7R immunoreactivity tend to have a poor prognosis in terms of disease-free survival. (**A**) Immunohistochemistry for IL-7R was performed with human ESCC tissues. The left and right images show low and high IL-7R intensities in a cancer nest, respectively. Scale bar: 20 µm (400×). (**B**) Correlation between IL-7R expression in ESCC tissues and ESCC patient prognosis. ESCC patients were divided into two groups according to the immunohistochemical intensity of IL-7R: the low-intensity group (*n* = 20) and high-intensity group (*n* = 48). Kaplan–Meier analysis of ESCC patients with low and high IL-7R expression was performed for overall, disease-free, and cause-specific survival. Data were analyzed using the log-rank test.

**Table 1 cancers-15-00394-t001:** Relationship between IL-7R expression levels and clinicopathological factors in human esophageal squamous cell carcinoma tissues.

		Expression of IL-7R ^a^		
	Number	Low (*n* = 20)	High (*n* = 49)	*p*-Value
Age				
<65	32	10	22	0.700
≥65	37	10	27	
Sex				
Male	55	19	36	0.044 *
Female	14	1	13	
Histological grade ^b^				
HGIEN + WDSCC	15	7	8	0.088
MDSCC + PDSCC	54	13	41	
Depth of tumor invasion ^b^				
T1	48	17	31	0.075
T2 + T3	21	3	18	
Lymphatic vessel invasion ^b^				
Negative	37	14	23	0.081
Positive	32	6	26	
Blood vessel invasion ^b^				
Negative	43	16	27	0.053
Positive	26	4	22	
Lymph node metastasis ^b^				
Negative	43	16	27	0.053
Positive	26	4	22	
Stage ^c^				
0 + I	38	14	24	0.111
II + III + IV	31	6	25	
Expression of αSMA ^d^				
Low	36	15	21	0.015 *
High	33	5	28	
Expression of FAP ^d^				
Low	39	15	24	0.048 *
High	30	5	25	
Expression of CD68 ^e^				
Low	35	14	21	0.041 *
High	34	6	28	
Expression of CD163 ^e^				
Low	34	14	20	0.028 *
High	35	6	29	
Expression of CD204 ^e^				
Low	34	15	19	0.006 **
High	35	5	30	

Data were analyzed using the *χ^2^*-test; *p* < 0.05 indicated statistically significant differences: ** p* < 0.05, *** p* < 0.01. ^a^ Based on the immunohistochemical intensity of IL-7R in the tumor nests, the human ESCC samples were divided into two groups: high-intensity group (High) and low-intensity group (Low). ^b^ Based on the 10th edition of the Japanese Classification of Esophageal Cancer [24]: HGIEN, high-grade intraepithelial neoplasia; WDSCC, well differentiated squamous cell carcinoma; MDSCC, moderately differentiated squamous cell carcinoma; PDSCC, poorly differentiated squamous cell carcinoma. T1, tumor invades from the superficial layer to the submucosa; T2, tumor invades the muscularis propria; T3, tumor invades the adventitia. ^c^ Based on the 7th edition of TNM classification by UICC [25]. ^d^ Immunoreactivity in the vicinity of the tumor’s invasive front was used to categorize patients into low- and high-groups. The cut-off value was set at 30% (high: >30%; low: ≤30%) [26]. ^e^ The median values of CD68-, CD163-, or CD204-positive macrophages in the tumor nests and stroma area was calculated. The patients were divided into low and high groups using the median value [19].

## Data Availability

The data presented in this study are available on request from the corresponding author.

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
