# Peer review of "Roles of IL-7R Induced by Interactions between Cancer Cells and Macrophages in the Progression of Esophageal Squamous Cell Carcinoma"

_cancers, 2023, doi:10.3390/cancers15020394_

Round 1

Reviewer 1 Report (Previous Reviewer 2)

The authors have performed additional experiments to address the reviewer's concerns and have revised the manuscript accordingly and extensively. Although some language issues still remain, I am ready to accept them.  

Author Response

Response to Reviewer 1 Comments

The authors have performed additional experiments to address the reviewer's concerns and have revised the manuscript accordingly and extensively. Although some language issues still remain, I am ready to accept them.  

→Thank you for your comments. We have thoroughly edited the manuscript for enhanced language and grammar and improved clarity.

Reviewer 2 Report (Previous Reviewer 1)

Thanks for the corrections.

Author Response

Response to Reviewer 2 Comments

Thanks for the corrections.

→Your comments have greatly helped me to improve the quality of my paper. Thank you very much.

This manuscript is a resubmission of an earlier submission. The following is a list of the peer review reports and author responses from that submission.

Round 1

Reviewer 1 Report

The authors found IL-7-IL7R interaction was involved in ESCC progression, IL-7R signals was showed to activate Akt and Erk1/2 signal, and promote the cancer cell migration. IL-7R expression was associated to poor clinical course. These findings are new aspect in this field, and impressive.

Comments;

#1; It was not clear why the authors focused on IL-7 in this study. Reviwer cannot see Table S1 in the web system, please check.

#2; In the figure 1, IL-7 protein was detected in co-culture medium?

#3; The methods of survival assay were not clear. Additional scheme might be useful for understanding of readers.

#4; Introduction; “M1/M2” should be changed to “M1/M2-like” (please see PMID: 33905102).

Reviewer 2 Report

The study assessed the effect of the co-culture of ESCC cells with macrophages on cancer cell proliferation, migration, and survival and the signaling pathways involved. It was found that IL-7R was highly expressed in ESCC along with Erk1/2 and Akt phosphorylation. Knockdown of IL-7R or inhibition of Erk or Akt signaling interfered with cell proliferation, migration, and survival. Overall, the data are sound and the manuscript was written fairly well. However, several concerns are needed to be addressed before publishing:

1.      A rationale is needed for every experiment that was done. For instance, why were Erk and Akt selected to examine, not JNK, NFκB, or any other pathways? Why was IL-7R selected to study, not other IL receptors? etc.

2.      The authors concluded that IL-7R promoted cell survival and growth in an IL-7-independent manner. In order to make this statement, IL-7 knockdown or inhibition experiments are required.

3.      The authors found that the knockdown of IL-7R reduced cell growth and survival. Did the authors try to determine in what way these events took place? Apoptosis, necroptosis, autophagy, or other types of cell death?

4.      Some contrastive connectives, such as “however”, “therefore”, etc., were misused throughout the paper. A tune-up in English is recommended.